**Data Availability Statement:** Data cannot be shared publicly because of limitations in the ethical

# COVID-19-related stigma among infected people in Sweden; psychometric properties and levels of stigma in two cohorts as measured by a COVID-19 stigma scale

Maria Reinius[1], Veronica Svedhem[2,3], Judith Bruchfeld[4,5], Heidi Holmström Larm[2], Malin Nygren-Bonnier[6,7], Lars E. Eriksson [2,6,8]*

1 Medical Management Centre, Department of Learning, Informatics, Management and Ethics, Karolinska Institutet, Stockholm, Sweden, 2 Medical Unit Infectious Diseases, Karolinska University Hospital, Huddinge, Sweden, 3 Unit of Infectious Diseases, Department of Medicine Huddinge, Karolinska Institutet, Huddinge, Sweden, 4 Department of Infectious Diseases, Karolinska University Hospital, Stockholm, Sweden, 5 Division of Infectious Diseases, Department of Medicine Solna, Karolinska Institutet, Stockholm, Sweden, 6 Department of Neurobiology, Care Sciences and Society, Karolinska Institutet, Huddinge, Sweden, 7 Medical Unit Occupational Therapy and Physiotherapy, Women's Health and Allied Health Professionals Theme, Karolinska University Hospital, Huddinge, Sweden, 8 School of Health and Psychological Sciences, City, University of London, London, United Kingdom

* lars.eriksson@ki.se

## Abstract

### Background

Epidemics have historically been accompanied by stigma and discrimination. Disease-related stigma has often been shown to have severe consequences for physical, mental and social wellbeing and lead to barriers to diagnosis, treatment and prevention. The aims of this study were to investigate if a HIV-related stigma measure could be adapted and valid and reliable to measure COVID-19-related stigma, and also to investigate levels of self-reported stigma and related factors among people in Sweden with experience of COVID-19 and compare levels of COVID-19-related stigma versus HIV-related stigma among persons living with HIV who had experienced a COVID-19 event.

### Methods

Cognitive interviews (n = 11) and cross-sectional surveys were made after the acute phase of the illness using a new 12-item COVID-19 Stigma Scale and the established 12-item HIV Stigma Scale in two cohorts (people who had experienced COVID-19 (n = 166/209, 79%) and people living with HIV who had experienced a COVID-19 event (n = 50/91, 55%). Psychometric analysis of the COVID-19 Stigma Scale was performed by calculating floor and ceiling effects, Cronbach's α and exploratory factor analysis. Levels of COVID-19 stigma between groups were analysed using the Mann-Whitney U test. Levels of COVID-19 and HIV stigma among people living with HIV with a COVID-19 event were compared using the Wilcoxon signed-rank test.

approval. We are unable to share data sets due to GDPR restrictions in Sweden and the EU. Data is available upon reasonable request to the head of department, Department of Neurobiology, Care Sciences and Society, 23400, Karolinska Institutet, SE-141 83 Huddinge, Sweden; email: prefekt@nvs.ki.se.

**Funding:** LEE, Swedish Research Council, grant no 2019-01222, https://www.vr.se/english.html LEE, The Strategic Research Area Healthcare Sciences, Karolinska Institutet, https://ki.se/en/research/about-sfo-v The funders had no role in study design, data collection and analysis, decision to publish, or preparation of the manuscript.

**Competing interests:** The authors have declared that no competing interests exist.

## Results

The COVID-19 cohort consisted of 88 (53%) men and 78 (47%) women, mean age 51 (19–80); 143 (87%) living in a higher and 22 (13%) in a lower income area. The HIV + COVID-19 cohort consisted of 34 (68%) men and 16 (32%) women, mean age 51 (26–79); 20 (40%) living in a higher and 30 (60%) in a lower income area. The cognitive interviews showed that the stigma items were easy to understand. Factor analysis suggested a four-factor solution accounting for 77% of the total variance. There were no cross loadings, but two items loaded on factors differing from the original scale. All subscales had acceptable internal consistency, showed high floor and no ceiling effects. There was no statistically significant difference between COVID-19 stigma scores between the two cohorts or between genders. People living in lower income areas reported more negative self-image and concerns about public attitudes related to COVID-19 than people in higher income areas (median score 3 vs 3 and 4 vs 3 on a scale from 3–12, $Z = -1.980$, $p = 0.048$ and $Z = -2.023$, $p = 0.024$, respectively). People from the HIV + COVID-19 cohort reported more HIV than COVID-19 stigma.

## Conclusions

The adapted 12-item COVID-19 Stigma Scale may be valid and reliable for measurement of COVID-19-related stigma. However, specific items may need to be rephrased or replaced to better correspond to the COVID-19 context. People who had experienced COVID-19 reported low levels of COVID-19-related stigma in general but people from lower income areas had higher levels of negative self-image and concerns about public attitudes related to COVID-19 than people from areas with higher income, which may call for targeted interventions. Although exhibiting more pronounced HIV stigma levels, people living with HIV who had experienced COVID-19 reported COVID-19-related stigma of the same low magnitude as their peers not living with HIV.

## Background

In 2020, coronavirus disease 2019 (COVID-19) started to spread across the world and became the first global pandemic from a new air-born virus in over 100 years. In response, over 120 countries went into lockdown, but the new virus spread at an unprecedented rate [1] and almost three years later the number of confirmed cases reported to the World Health Organization (WHO) was almost 650 million with over 6.6 million deaths (covid19.who.int, accessed 11 December, 2022). The pandemic has had dire societal consequences, including economic crisis, and has substantially affected the lives of most of the global population [1]. Many scholars have also raised warnings that stigma related to COVID-19 could possibly have devastating effects on the global response [2–6] and the United Nations Children's Fund (UNICEF), the WHO, the International Federation of Red Cross and Red Crescent Societies (IFRC) and the US Centers for Disease Control and Prevention (CDC) have published guides concerning how to prevent stigma related to COVID-19 [7, 8].

Epidemics have historically been accompanied by stigma and discrimination, and disease-related stigma, here with examples from the human immunodeficiency virus (HIV) area, has often been shown to have severe consequences for the physical, mental and social aspects of people's health-related quality of life [9–11] and to lead to barriers to diagnosis, treatment and

prevention [12]; concerns have therefore been raised that this could also occur in the COVID-19 pandemic [7]. People are often afraid of the unknown and the COVID-19 pandemic may have led to confusion, anxiety and fear among the public [7]. These factors may in turn fuel the development of harmful stereotypes [7], for example that some groups of people are more likely than others to spread the virus [8]. The social mechanism of labelling people based on stereotypical beliefs is a core foundation of social stigma, as described by Goffman [13]. People who are labelled and stereotyped often experience status loss and discrimination, and the whole process of stigmatization is contingent to unequal distributions of social, economic and political power [14].

It has been reported that stigma related to COVID-19 leads to people being reluctant to disclose their COVID-19 diagnosis and experiencing telling someone as a risk [15]; qualitative studies confirm this [16]. For example, participants in a study of how people with COVID-19 experienced hospitalization expressed initial fear and feelings of stigma and discrimination. Some also considered concealing their contact history when being admitted to and discharged from hospital [16]. Data from India obtained from both COVID-19 recovered and non-COVID-19 infected individuals show that half of the non-COVID-19 infected participants reported severe stigmatizing attitudes towards COVID-19 infected persons, while 40% of COVID-19 recovered participants reported experiencing severe stigma [17]. High levels of stigma towards people with ongoing COVID-19 or those who have recovered were confirmed in a recent study among the Egyptian general population [18]. Furthermore, a study including back-to-school students in Wuhan, China after the initial wave of transmission showed that discrimination, internalized stigma and shame was associated with negative mental health outcomes [19]. A study on COVID-19-related stigma and mental health among healthcare workers in Vietnam suggests that the dominant areas may be negative self-image and concerns about public attitudes related to COVID-19 [20]. Although the sample size was small (n = 61) and stigma related to COVID-19 was measured with an instrument not extensively validated, the findings indicate that healthcare workers may feel guilt and avoid contact with others. A larger Egyptian study has confirmed high levels of experienced healthcare worker related stigma in physicians in the context of the COVID-19 pandemic [21].

In many countries people have reported reduced access to HIV care during the COVID-19 pandemic [22, 23]. HIV care and test facilities were closed down due to healthcare personnel needing to focus on the treatment of patients with COVID-19, making it more difficult for some to access antiretroviral treatment [22, 24]. Concerns have also been raised that stigma related to COVID-19 could contribute to layered stigma for people living with HIV [23]. A symposium at the 2021 International AIDS Society Conference drew parallels between the HIV and COVID-19 pandemics and stated that healthcare systems now need to adopt a long-term approach to COVID-19, looking at patients' health and wellbeing in a holistic way [24]. A prerequisite for tracking progress in patients' health and wellbeing is the availability of valid and reliable instruments; however, there is currently a lack of knowledge about the measurement of stigma related to COVID-19.

More research is needed concerning how stigma related to COVID-19 can be assessed in a reliable and valid way, and cross-sectional studies are needed to examine which groups experience stigma related to COVID-19 and in what contexts and situations, as well as the relationship of such stigma to other health outcomes. The purpose of this study was, therefore, to provide knowledge about how stigma related to COVID-19 can be measured using valid and reliable methods. The Berger HIV stigma scale [25] is a commonly used measure for assessing HIV stigma. A review has recently investigated the psychometric properties of different variants of the Berger HIV stigma scale in 166 scientific articles and found the instrument to be valid and reliable for the measurement of HIV-related stigma in different contexts [26].

Building on knowledge about HIV-related stigma, the present study also aims to expand the understanding of stigma related to COVID-19 by comparing experiences of stigma related to HIV and COVID-19, respectively. The specific aims of the study were to 1. Investigate if an instrument for the measurement of HIV-related stigma could be adapted and be valid and reliable to measure COVID-19-related stigma, 2. To investigate the levels of self-reported stigma and related factors among people in Sweden diagnosed with COVID-19, and 3. To compare the levels of COVID-19-related stigma versus HIV-related stigma among persons living with HIV who had experienced a COVID-19 event.

## Methods

A cross-sectional survey was performed with two cohorts: people who had experienced COVID-19 and people living with HIV who had experienced a COVID-19 event.

### Ethical approval

This study was approved by the Swedish Ethical Review Agency (Dnr 2020–04242).

### Context

The first confirmed cases of COVID-19 in Sweden occurred in January 2020 among people coming home from travels abroad and people residing in the capital of Sweden. The first outbreak may have been related to a school winter break in March 2020 when many families travelled abroad for vacation. The Swedish COVID-19 response was initially criticized both nationally and internationally [27] but has since been highlighted in international media as being potentially beneficial in the long run [28]. Qualitative studies have found that members of the Swedish public expressed strong support for the Swedish pandemic response [29].

### Measurements

**The 12-item COVID-19 Stigma Scale.** The COVID-19 Stigma Scale was adapted by the research group from the 12-item HIV Stigma Scale [30], previously developed from Berger HIV Stigma Scale [25] and validated for use in a Swedish context [30]. A recent review found the Berger HIV Stigma Scale to be valid and reliable for measurement of HIV-related stigma in different contexts [26]. The 12-item HIV Stigma Scale includes four subscales with three items in each (Personalized stigma, Disclosure concerns, Negative self-image, and Concerns about public attitudes). The items are answered on a 4-point Likert scale from strongly disagree (1) to strongly agree (4) and the three items for each subscale are summed to form the subscale score with a possible range from 3 to 12 –the higher the score, the higher the rated stigma [30]. The scale was adapted for COVID-19-related stigma by consistently changing "HIV" to "COVID-19" for each item. Some items were also rephrased to better fit the context of COVID-19. For example, the tempus was altered in some questions from "having HIV" to "having had COVID-19". The survey also contained questions about whether the participants had received any in-hospital care in connection with their COVID-19 and the following data were retrieved from participants' medical records: age, gender, postal code, date of positive COVID-19 test, hospital care (yes/no and length of stay), intensive care (yes/no and length of stay). Postal codes were manually converted to Regional Statistical codes (RegSo) and mean incomes (in 2019) for each RegSo were retrieved from Statistics Sweden [31]. Mean income by RegSo was then categorized into above or below the mean income in Sweden (326 000 SEK/year in 2019).

**The 12-item HIV Stigma Scale.** The participants from the HIV + COVID-19 cohort (see below) were also asked to answer the 12-item HIV Stigma Scale [30] described above.

## Participants and procedures

Aspects of validity of the adapted COVID-19 Stigma Scale were tested using think-aloud cognitive interviews. Survey data were collected from two cohorts (see below) and formed the basis for psychometric analysis of the COVID-19 stigma scale, and for descriptive and comparative analyses regarding COVID-19-related and HIV-related stigma.

**Cognitive interviews.** Think-aloud cognitive interviews [32] were performed in parallel to the cross-sectional survey in order to assess the validity of the items of the 12-item COVID-19 Stigma Scale. Eleven individuals who had had COVID-19 (7 women, 4 men; aged between 26 and 78 years; 8 with Swedish, 1 with Southern European and 2 with African ancestry) were identified and purposively selected through a Swedish infectious disease clinic. They were invited to complete the questionnaire whilst speaking their thoughts out loud in individual interviews with a research assistant. The research assistant made field notes about the participants thoughts and reactions. After the questionnaire had been completed, the research assistant asked supplementary questions from an interview guide to gain a deeper understanding of the particular questions the participant had commented on during the think-aloud phase of the interview. Overall questions were also asked about the relevance of the questionnaire for people with COVID-19 and how it felt to complete the questionnaire.

**Survey.** Survey data using the 12-item COVID-19 Stigma Scale were collected from two cohorts, in this article referred to as 1) The COVID-19 cohort and 2) The HIV + COVID-19 cohort.

**The COVID-19 cohort** includes people living in Sweden who travelled abroad in February 2020 and came back to Sweden with either ongoing COVID-19 or experienced the onset of their COVID-19 in close connection to their return to Sweden. Within about two to three months after their diagnosis, the COVID-19 Stigma Scale was distributed to 209 individuals and 167 answered and returned the scale, of which 166 had valid responses (response rate 79%).

**The HIV + COVID-19 cohort** includes people living with HIV in Sweden and who had been diagnosed with COVID-19 between May 2020 and July 2021. Ninety-one people living with HIV with positive serology for COVID-19 were consecutively asked to answer both the COVID-19 stigma and HIV stigma scales a period after their COVID-19 event (the participants were free to choose the order in which to answer the two scales); 50 (55%) responded.

## Analysis

Statistical analyses were performed with IBM SPSS Statistics version 28.

**Interviews.** Written records from the cognitive interviews were analysed by summarising the comments and remarks item by item, forming part of the validity assessment.

**Psychometric analysis of the COVID-19 Stigma Scale.** The combined sample of the two cohorts was used for the psychometric analysis. Floor and ceiling effects at subscale level were calculated to assess the validity of the scale. Reliability (internal consistency) was assessed through Cronbach's $\alpha$; $\alpha$ over 0.7 was deemed acceptable [33]. We also calculated $\alpha$ if item deleted. An exploratory factor analysis (EFA) was performed with $\alpha$ factoring, oblimin rotation. Missing answers were handled using listwise deletion (for each subscale separately) in the psychometric analysis.

**Comparison of levels of stigma across and within cohorts.** Some participants had omitted to give responses to items on the 12-item COVID-19 Stigma Scale and/or the 12-item HIV

Stigma Scale. In order to further enable comparisons across and within cohorts after completion of the psychometric analysis we used an imputation algorithm as follows. If a participant had not answered one of the three items in a COVID-19 stigma or HIV-stigma subscale, the missing value was imputed with a random of that participant's responses to the two remaining items of that scale. If a participant had not answered two of the three items in a subscale, the missing values were imputed with that participant's single response to the remaining item in that subscale. If answers were missing for all three items in a subscale, values were not imputed and the participant was excluded from that part of the analysis. In total, fifteen and two imputations were made for the COVID-19 Stigma Scale and the HIV Stigma Scale, respectively.

Potential differences in background data between the two cohorts were analysed by $\chi^2$ tests. Levels of stigma related to COVID-19 were compared between the two cohorts, between men and women, between lower and higher income municipalities, and between those participants that had been hospitalised and those not hospitalised in connection to their COVID-19 event using the Mann-Whitney U test. Levels of stigma related to COVID-19 and HIV respectively were compared for the HIV + COVID-19 cohort using the Wilcoxon signed-rank test. Missing scores were handled using casewise deletion in the comparisons of stigma levels. P-values below 0.05 were deemed statistically significant.

## Results

Sociodemographic information for the two cohorts is presented in Table 1. There were no statistically significant differences between the cohorts regarding gender, age and how many had

**Table 1. Characteristics of study participants.**

| Characteristics | COVID-19 cohort (n = 166) n (%) | HIV+cCOVID-19 cohort (n = 50) n (%) | $\chi^2$, p |
|---|---|---|---|
| **Gender** | | | 3.51, p = 0.061 |
| Male | 88 (53) | 34 (68) | |
| Female | 78 (47) | 16 (32) | |
| **Age (years)[a]** | | | 5.86, p = 0.134 |
| 18–20 | 7 (4) | 0 (0) | |
| 21–40 | 12 (7) | 8 (16) | |
| 41–60 | 120 (72) | 33 (66) | |
| 61–80 | 27 (16) | 9 (18) | |
| **Living in an area with mean income above or below national mean income** | | | 45.57, p<0.001 |
| Above mean income | 143 (87) | 20 (40) | |
| Below mean income | 22 (13) | 30 (60) | |
| **Missing answers** | 1 (1) | 0 (0) | |
| **Admitted to hospital** | | | 20.10, p<0.001 |
| Yes | 37 (22) | 3 (6) | |
| No | 122 (74) | 36 (72) | |
| Missing answer | 7 (4) | 11 (22) | |
| **Admitted to intensive care unit** | | | 1.47, p = 0.480 |
| Yes | 10 (6) | 1 (2) | |
| No | 151 (91) | 48 (96) | |
| Missing answer | 5 (3) | 1 (2) | |

[a]mean age 51 (19–80) and 51 (26–79) years for the COVID-19 and HIV + COVID-19 cohorts respectively, t = -0.120, p = 0.904 (independent samples t-test).

been admitted to intensive care units related to COVID-19. Participants in the COVID-19 cohort were more likely to live in an area with higher income and have been admitted to hospital related to COVID-19 than participants in the HIV + COVID-19 cohort ($\chi^2$ = 45.57, $p<0.001$ and $\chi^2$ = 20.10, $p<0.001$, respectively).

## Validity

The cognitive interviews showed that the COVID-19 stigma items were easy to understand, in general. However, participants asked about definitions for "worse" in relation to the item *People's attitudes about COVID-19 make me feel worse about myself* and wondered about the word risky in the item *Telling someone I have COVID-19 is risky* (subscale Disclosure concerns) and asked "risky for who?". One participant reacted to the latter item and said that it was the opposite, that people became relieved when hearing that someone had been through COVID-19. Some participants found questions on the subscale "Concerns about public attitudes" difficult to answer and said that it was hard to know about the situation for people with COVID-19 in general. Participants did not know if they should answer according to their own experience or what they thought about the situation in general. Some participants found the item *Some people avoid touching me once they know I have had COVID-19* (subscale Personalized stigma) difficult to answer because everyone avoids physical contact these days. To a direct question about whether the items were relevant in general, some participants responded that the items could be relevant for people with COVID-19 but were not relevant for themselves. They said that they did not feel exposed or experience that they were an outcast because of COVID-19 and their point of view was that COVID-19 was nothing special. One participant became upset by reading the question *I feel guilty because I have had COVID-19* (subscale Negative self-image) and said "why [offensive word] should I feel that?". Participants also reflected on time aspects and said that people could be treated a bit like an outcast if they had an ongoing infection but not when you had recovered from the infection, when it could be the opposite. COVID-19 was also said to be considered tabu at the beginning of the pandemic but not so much anymore. Some participants thought that it could be relevant to add questions about attitudes among healthcare personnel and about how people with COVID-19 were treated by healthcare professionals.

Mean scores, and floor and ceiling effects of the COVID-19 stigma subscales from the cross-sectional survey are shown in Table 2. All subscales showed high floor effects and no ceiling effects, since a large proportion (54–71%) of participants answered with the lowest possible scores (totally disagree) on all items in a subscale.

## Construct validity

The dataset with the two cohorts combined (n = 200 participants who completed all 12 COVID-19 stigma items) was found suitable for factor analysis with a Kaiser-Meyer-Olkin measure of sampling adequacy of 0.871 and a $p$-value below 0.001 for Bartlett's test of sphericity. A scree plot suggested a four-factor solution but only three factors had an eigenvalue above one. Four factors accounted for 77% of the total variance. There were no cross loadings, but two items loaded on factors that were not expected (see Table 3); the item *I feel I'm not as good a person as others because I have had COVID-19* loaded on Personalized stigma instead of Negative self-image and the item *Some people avoid touching me once they know I have had COVID-19* loaded on Concerns about public attitudes instead of Personalized stigma.

## Reliability

Cronbach's α and α if item deleted are shown in Table 2. All subscales had acceptable internal consistency (Cronbach's α >0.7). Analysis of α if item deleted indicated that α would improve for

**Table 2. COVID-19 stigma scale mean scores and psychometric properties.**

| | N complete answers* | Mean item score (range 1–4) | Mean subscale score (range 3–12) | Reliability (a) | Alpha if item deleted | Floor/ceiling effect (%) |
|---|---|---|---|---|---|---|
| **Personalized stigma** | 213 | | 3.79 | 0.750 | | 66/1 |
| Some people avoid touching me once they know I have had COVID-19 | | 1.47 | | | 0.848 | |
| People I care about stopped calling after learning I have had COVID-19 | | 1.17 | | | 0.528 | |
| I have lost friends by telling them I have had COVID-19 | | 1.16 | | | 0.681 | |
| **Disclosure concerns** | 211 | | 4.09 | 0.844 | | 60/1 |
| Telling someone I have had COVID-19 is risky | | 1.40 | | | 0.818 | |
| I work hard to keep my COVID-19 diagnosis a secret | | 1.24 | | | 0.779 | |
| I am very careful about who I tell that I have had COVID-19 | | 1.44 | | | 0.753 | |
| **Negative self-image** | 215 | | 3.61 | 0.743 | | 71/0 |
| I feel guilty because I have had COVID-19 | | 1.33 | | | 0.648 | |
| People's attitudes about COVID-19 make me feel worse about myself | | 1.19 | | | 0.608 | |
| I feel I'm not as good a person as others because COVID-19 | | 1.12 | | | 0.706 | |
| **Concerns with public attitudes** | 210 | | 4.13 | 0.796 | | 54/1 |
| People with COVID-19 are treated like outcasts | | 1.42 | | | 0.765 | |
| Most people believe a person who has had COVID-19 is dirty | | 1.25 | | | 0.704 | |
| Most people are uncomfortable around someone who has had COVID-19 | | 1.47 | | | 0.701 | |

*missing answers omitted listwise for each subscale separately.

the subscale Personalized stigma if the item *Some people avoid touching me once they know I have COVID-19* was deleted (α 0.750 with the item included vs. α 0.848 with the item deleted).

**Differences in stigma experiences.** There was no statistically significant difference between COVID-19 stigma scores reported by the two cohorts, between men and women (Table 4) or between those who had been hospitalised and those who had not been hospitalised in connection to their COVID-19 event (data not shown). In the comparison between participants from a lower or a higher-income municipality, there were no differences on the COVID-19 stigma scores for Personalized stigma or Disclosure concerns (see Table 4). For Negative self-image and Concerns about public attitudes there were statistically significant differences where people residing in lower income areas reported more Negative self-image and Concerns about public attitudes than people residing in higher income areas (median score 3 vs 3 and 4 vs 3 on a scale from 3–12, Z = -1.980, asymp. Sig. (two-tailed) = 0.048 and Z = -2.023, asymp. Sig.(two-tailed) = 0.043; Table 4). People from the HIV + COVID-19 cohort reported more stigma related to HIV than to COVID-19 and this difference was statistically significant for all four subscales (Table 5).

## Discussion

This study aimed to investigate if the 12-item HIV stigma scale could be adapted and then be valid and reliable for the measurement of stigma related to COVID-19; it also examined levels

**Table 3. Pattern matrix for exploratory factor analysis (alpha factoring, oblimin rotation).** Factor loadings <0.32 not shown. N = 200.

| Item | Factor 1 (Personalized stigma) | Factor 2 (Negative self-image) | Factor 3 (Disclosure concerns) | Factor 4 (Concerns about public attitudes |
|---|---|---|---|---|
| I have lost friends by telling them I have had COVID-19 | 0.874 | | | |
| People I care about stopped calling after learning I have had COVID-19 | 0.724 | | | |
| *I feel I'm not as good a person as others because COVID-19\** | *0.563\** | | | |
| People's attitudes about COVID-19 make me feel worse about myself | | 0.805 | | |
| I feel guilty because I have had COVID-19 | | 0.574 | | |
| I work hard to keep my COVID-19 diagnosis a secret | | | 0.901 | |
| I am very careful about who I tell that I have had COVID-19 | | | 0.734 | |
| Telling someone I have had COVID-19 is risky | | | 0.674 | |
| Most people believe a person who has had COVID-19 is dirty | | | | 0.797 |
| Most people are uncomfortable around someone who has had COVID-19 | | | | 0.684 |
| *Some people avoid touching me once they know I have had COVID-19\** | | | | *0.611\** |
| People with COVID-19 are treated like outcasts | | | | 0.539 |

*Items in italic = Items that load on an unexpected factor.

of self-reported stigma and associated factors among people in Sweden diagnosed with COVID-19. Data were collected from two cohorts, one consisting of people with experience of having had COVID-19 in the earliest wave of the pandemic and one of people living with HIV who had also experienced a COVID-19 event.

**Table 4. Comparison of COVID-19 stigma levels between different groups, Mann-Whitney U test.**

| | Cohort | | | Gender (combined cohorts, n = 216) | | | Residential area income level (combined cohorts, n = 216) | | |
|---|---|---|---|---|---|---|---|---|---|
| | COVID-19[a] | HIV+COVID-19[b] | | Men[d] | Women[e] | | Lower[f] | Higher[g] | |
| | $\bar{x}$/M | $\bar{x}$/M | U (p)[c] | $\bar{x}$/M | $\bar{x}$/M | U (p)[c] | $\bar{x}$/M | $\bar{x}$/M | U (p)[c] |
| Personalized stigma COVID-19 | 3.78/3 | 3.92/3 | 3762 (0.488) | 3.81/3 | 3.82/3 | 5513 (0.740) | 4.02/3 | 3.75/3 | 3596 (0.100) |
| Disclosure concerns COVID-19 | 4.14/3 | 3.88/3 | 3710 (0.195) | 3.94/3 | 4.27/3 | 5125 (0.127) | 4.52/3 | 3.95/3 | 3801 (0.203) |
| Negative self-image COVID-19 | 3.58/3 | 3.86/3 | 3848 (0.325) | 3.69/3 | 3.60/3 | 5710 (0.948) | 4.12/3 | 3.50/3 | 3624 (**0.048\***) |
| Concerns about public attitudes COVID-19 | 4.16/3 | 4.18/3.5 | 3911 (0.498) | 4.16/3 | 4.16/3 | 5721 (0.975) | 4.67/4 | 4.01/3 | 3520 (**0.043\***) |

[a]n = 166

[b]n = 48–50

[c]Mann-Whitney U test, asymp. Significance

[d]n = 120–122

[e]n = 94

[f]n = 51–52

[g]n = 162–163; $\bar{x}$ mean; M median

*statistically significant, $p<0.05$.

**Table 5. Comparison between stigma scores related to COVID-19 and HIV in the cohort of people living with HIV who had experienced a COVID-19 event (n = 48[a]), Wilcoxon signed-rank test.**

| | People living with HIV | | | |
| --- | --- | --- | --- | --- |
| | (mean) | (median) | Z | Asymp. Sig *p* (two-tailed) |
| **Personalized stigma COVID-19[b]** | 3.80 | 3 | -4.113 | <0.001 |
| **Personalized stigma HIV[b]** | 5.80 | 6 | | |
| **Disclosure concerns COVID-19** | 3.77 | 3 | -5.797 | <0.001 |
| **Disclosure concerns HIV** | 9.42 | 10 | | |
| **Negative self-image COVID-19** | 3.79 | 3 | -5.014 | <0.001 |
| **Negative self-image HIV** | 6.58 | 6 | | |
| **Concerns about public attitudes COVID-19** | 4.12 | 3 | -5.675 | <0.001 |
| **Concerns about public attitudes HIV** | 7.46 | 7 | | |

[a]The HIV + COVID-19 cohort constituted 50 cases but 2 had missing responses on the HIV stigma scale and were therefore excluded.

We found support for reliability and construct validity of the scale but also evidence suggesting that some items were not optimally constructed for the specific context of COVID-19. Items indicating that people experience others avoiding physical contact with them have been found relevant for people living with HIV [25] and recently among physicians working during the COVID-19 pandemic [34]; however, results of the cognitive interviews in the present study indicate that the context of oneself having experienced a COVID-19 event is different regarding aspects of physical contact. Since the current COVID-19 situation results in people generally avoiding physical contact with others, it is plausible that people with COVID-19 do not experience stigma in the event of people avoiding physical contact with them. The psychometric results (α if item deleted and factor analysis) also support the suggestion that the item regarding physical contact can be omitted from a COVID-19-stigma scale or, as suggested by Mlouki et al [15], rephrased to better mirror the COVID-19 context.

Although there were internal differences depending on whether the participants lived in a lower or higher income area, the present study indicates that people in Sweden who have had COVID-19 do not generally report experiencing any pronounced levels of stigma related to their COVID-19. Floor effects were high in both cohorts, meaning that most participants answered with the lowest possible score ("totally disagree") to each of the items in the respective subscale. This stands in contrast to reports of exhibited COVID-19 stigma in other countries around the world [2]. In addition, a qualitative study in Sweden's neighbouring country Finland has shown that people with COVID-19 perceived stigma, disclosure concerns and self-stigma related to the virus [35].

Time aspects need to be considered in the measurement of COVID-19-related stigma. In the present study, surveyed participants had passed the initial symptomatic phase of the disease and both cognitive interviews and psychometric analysis indicated that the participants did not experience COVID-19-related stigma at the time of answering the questionnaire. However, all participants in the cognitive interviews stated that the questions could be relevant for other people with COVID-19, with a possible interpretation being that this refers to persons with ongoing infection. Earlier qualitative studies support the suggestion that people with COVID-19 experience stigma and own negative feelings towards the disease at the beginning of their infection, but that these feelings gradually shifted towards a mix of positive and negative feelings [16]. Further research is needed to explore the time aspects of experiences of stigma related to COVID-19. In the cognitive interviews, participants also reflected on time aspects of the pandemic itself, where they thought that having COVID-19 was considered more tabu at the beginning of the pandemic. It is also likely that the time aspect explains why

participants from the COVID-19 cohort were admitted to hospital more frequently than participants from the HIV + COVID-19 cohort; the routines for treatment and criteria for in-hospital treatment have developed since the very first waves of the pandemic. Respiratory symptoms became possible to treat in outpatient care to an increased extent and inpatient care was foremost indicated for those with comorbidities and other risk factors, such as high age.

Severity of symptoms could be suspected to influence affected peoples' experiences from going through a COVID-19 event. In the present study, we did not have any details about symptomatology of the participants although hospitalisation status could be assumed to be a marker of presence of severe symptoms. When comparing the group of participants that had been hospitalised in relation to their COVID-19 event with those who had not, similar to the results recently presented by Avila et al [36], hospitalisation status did not show any differences in relation to experiences of COVID-19-related stigma.

In this study, people living in municipalities with lower income levels reported statistically significantly higher levels of negative self-image and concerns about public attitudes due to COVID-19 compared to people living in municipalities with higher income levels. A previous Swedish study has shown that people living with HIV are less likely to be employed than corresponding people not living with HIV [37]. One can speculate that in municipalities where people are at higher risk of financial and other constraints (e.g. unemployment) that could be further endorsed by a stigmatised attribute, negative self-image because of COVID-19 and fear of stigma related to people's attitudes could be more pronounced as is also proposed by Imran et al [38] in a Pakistani context. Infectious disease-related stigma could lead to concealment and therefore an increased risk of further spread of disease [12]. The results of the present study, indicating higher levels of self-perceived stigma in deprived areas, call for interventions to prevent or mitigate stigma among populations residing in such areas to decrease associated suffering and reduce the risk of further spread of the infection. Further research is needed to explore the relationship between COVID-19 stigma and other mental health aspects of quality of life and whether there are groups at risk of stigmatisation and related negative effects on mental and physical health outcomes.

A cohort of people living with HIV completed both the COVID-19 Stigma Scale and the HIV Stigma Scale in this study. Their reported HIV stigma scores were in line with previous studies from Sweden [9, 30], however their reported HIV stigma scores were statistically significantly higher than their reported COVID-19 stigma scores. It has been suggested that people with HIV could experience more stigma related to COVID-19 than others [39], a feature that was not confirmed in this study. Further, a recent review on social and behavioural impacts of COVID-19 found that although people with HIV in some contexts had reduced access to antiretroviral therapy and the health service during the pandemic, there was no consistent support for the suggestion that people with HIV would experience increased stigma in relation to COVID-19. The authors of the review suggest that the experience of living with HIV could work as a source of resilience towards COVID-19-related stigma [23], which is also in line with the results of the present study which show no difference in levels of COVID-19 stigma between participants living or not living with HIV. Further, the authors of the review [23] suggest that people with the stigmatized condition HIV would have a sense of what stigma related to COVID-19 could be, and thereby, as supported by qualitative findings, people living with HIV may use their experience from the HIV pandemic to not stigmatize people with COVID-19 [40].

## Strengths and weaknesses

When this study was designed, we were not aware if participants would experience stigma related to having COVID-19. It is not ideal to test the validity of a scale among participants

who only have vague experiences of the phenomena the scale is intended to capture. However, the participants found that the items were relevant for other persons with a COVID-19 experience, although not for themselves in their own situation. Further psychometric testing is needed with participants who do experience stigma related to having COVID-19. The fact that the COVID-19 cohort was mainly collected from people experiencing COVID-19 during the first wave of the pandemic, while the HIV + COVID-19 cohort was collected over a longer timespan could be considered a weakness. One strength to be highlighted is the high response rate in the COVID-19 cohort (79%), indicating that the participants considered the topic of high relevance. A further strength is the mixed methods design of the evaluation of the COVID-19 Stigma Scale, using both qualitative and quantitative data.

## Conclusions

The 12-item COVID-19-adapted version of the 12-item HIV Stigma Scale may be valid and reliable for the measurement of COVID-19-related stigma. Specific items may need to be rephrased or replaced with items that better correspond to the context of COVID-19. People in Sweden who had experienced COVID-19 reported low levels of stigma related to COVID-19 in general, however people from areas with lower income reported significantly higher levels of negative self-image and concerns about public attitudes related to COVID-19 than people from areas with higher income which indicate the need of targeted efforts aiming to prevent suffering and further spread of the infection. Although exhibiting more pronounced HIV stigma levels, people living with HIV who had experienced a COVID-19 event, reported COVID-19 stigma of the same low magnitude as their peers not living with HIV.

## Acknowledgments

We acknowledge all study participants who so generously spent time and effort participating in the interviews and answering the survey.

## Author Contributions

**Conceptualization:** Maria Reinius, Veronica Svedhem, Judith Bruchfeld, Heidi Holmström Larm, Malin Nygren-Bonnier, Lars E. Eriksson.

**Data curation:** Maria Reinius, Veronica Svedhem, Lars E. Eriksson.

**Formal analysis:** Maria Reinius, Lars E. Eriksson.

**Funding acquisition:** Lars E. Eriksson.

**Investigation:** Maria Reinius, Veronica Svedhem, Heidi Holmström Larm, Malin Nygren-Bonnier, Lars E. Eriksson.

**Methodology:** Maria Reinius, Lars E. Eriksson.

**Project administration:** Maria Reinius, Veronica Svedhem, Heidi Holmström Larm, Lars E. Eriksson.

**Resources:** Veronica Svedhem, Lars E. Eriksson.

**Supervision:** Veronica Svedhem, Lars E. Eriksson.

**Validation:** Maria Reinius, Lars E. Eriksson.

**Writing – original draft:** Maria Reinius, Veronica Svedhem, Lars E. Eriksson.

**Writing – review & editing:** Maria Reinius, Veronica Svedhem, Judith Bruchfeld, Heidi Holmström Larm, Malin Nygren-Bonnier, Lars E. Eriksson.

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
