## [Decision Letter · Decision Letter 0]

27 Mar 2023

PONE-D-22-35422Covid-19-related stigma among infected people in Sweden; psychometric properties and levels of stigma in two cohorts as measured by a covid-19 stigma scalePLOS ONE

Dear Dr. Eriksson,

Thank you for submitting your manuscript to PLOS ONE. After careful consideration, we feel that it has merit but does not fully meet PLOS ONE’s publication criteria as it currently stands. Therefore, we invite you to submit a revised version of the manuscript that addresses the points raised during the review process.

We look forward to receiving your revised manuscript.

Kind regards,

Nabeel Al-Yateem, PhD

Academic Editor

PLOS ONE

Journal Requirements:

Reviewers' comments:

Reviewer's Responses to Questions

**Comments to the Author**

1. Is the manuscript technically sound, and do the data support the conclusions?

Reviewer #1: Yes

Reviewer #2: Yes

2. Has the statistical analysis been performed appropriately and rigorously? 

Reviewer #1: I Don't Know

Reviewer #2: Yes

3. Have the authors made all data underlying the findings in their manuscript fully available?

Reviewer #1: Yes

Reviewer #2: No

4. Is the manuscript presented in an intelligible fashion and written in standard English?

Reviewer #1: Yes

Reviewer #2: Yes

5. Review Comments to the Author

Reviewer #1: General

The authors should be consistent in writing “Covid-19”.It is sometimes mentioned starting in capital letter (e.g. line 38) and sometimes starting in small letter (e.g. line 33).

Introduction

Line 95: “qualitative studies confirm this” Reference of qualitative study should me mentioned in the same sentence (Reference number 16)

Cognitive interviews

Line 178-182

How did the authors select the participants for Think-aloud cognitive interviews?

Analysis

Line 205

“Written records from the cognitive interviews were analysed by MR who summarised comments and 206 remarks item by item, which formed part of the validity assessment.”

It’s better to mention who performed the analysis in authors contribution section not within the manuscript

Results

Table 1. Characteristics of study participants

Data presentation of age is better than “Year of birth”

Merging of the cells of the same variable (e.g. gender) is better in data presentation

Line 255-257

“To a direct question about whether the items were relevant in general, some participants responded that the items could be relevant for people with covid-19 but were not relevant for them”. What do you mean by “them”? People who recovered from Covid-19?

Table 2

Merging of the cells of the same variable (i.e.scales) is better in data presentation

Discussion

Line 374 “… although people with HIV in some contexts had reduced access to ART and the health service during the pandemic …”

What does ART stand for?

The following manuscript may be helpful in discussion:

Validity and Reliability of a COVID-19 Stigma Scale Using Exploratory and Confirmatory Factor Analysis in a Sample of Egyptian Physicians: E16-COVID19-S. A Mostafa, NS Mostafa, N Ismail. International Journal of Environmental Research and Public Health 18 (10), 5451

Reviewer #2: The study presents original research on the development of an instrument to measure COVID-19 related stigma. The cross-sectional assessments and respective analyses are performed to a high technical standard and are described in sufficient detail. The article is presented in an intelligible fashion and is written in standard English.

One major comment refers to the results that people who had experienced COVID-19 reported low levels of covid-19-related stigma. This is stated in the abstract and text and it is a finding of the study. However, it seems, as authors state in the Discussion, that it would be more plausible to expect people to experience stigma during their COVID-19 event and that this stigma might not persist after the resolution of the disease. As such, it would be important to state, in the abstract and elsewhere, that covid-19-related stigma was not measured during the symptomatic phase of the disease. Two related questions are: 1) Did some participants answer the questionnaire during their COVID-19 event? And if yes, did stigma scores differ according to when it was measured?, 2) Do authors have information about symptomatic infection? If yes, could scores be compared between the two groups: symptomatic vs not?

Page 3/line 36: “HIV who had experienced a concomitant covid-19.” Though the sentence is likely correct, it might be easier to read if “event” or “episode” was added after covid-19. Though COVID-19 is the official name of the disease (in upper case), the text “a concomitant covid-19” reads awkward (as it would read if it was another disease, such as pneumonia or flu). Other instances such as this one can be found throughout the text.

Page 8/line 142: “The study reported in this article included people in Sweden who had had covid-19.” It is not clear how this sentence is informative and how it relates to the context of COVID-19 in Sweden.

Page 10/line 199: Did people living with HIV respond to both instruments consecutively? Which one was answered first? Because the items are the same (except for the words HIV/COVID-19), could this have influenced their responses and therefore the results?

6. PLOS authors have the option to publish the peer review history of their article (what does this mean?). If published, this will include your full peer review and any attached files.

Reviewer #1: No

Reviewer #2: No

---

## [Author Response · Author response to Decision Letter 0]

10 May 2023

Responses to the Academic Editor's and the reviewers' comments is included in the attached "Response to Reviewers" document.

---

## [Decision Letter · Decision Letter 1]

5 Jun 2023

COVID-19-related stigma among infected people in Sweden; psychometric properties and levels of stigma in two cohorts as measured by a COVID-19 stigma scale

PONE-D-22-35422R1

Dear Dr. Eriksson,

We’re pleased to inform you that your manuscript has been judged scientifically suitable for publication and will be formally accepted for publication once it meets all outstanding technical requirements.

Kind regards,

Nabeel Al-Yateem, PhD

Academic Editor

PLOS ONE

Additional Editor Comments (optional):

Reviewers' comments:

Reviewer's Responses to Questions

**Comments to the Author**

1. If the authors have adequately addressed your comments raised in a previous round of review and you feel that this manuscript is now acceptable for publication, you may indicate that here to bypass the “Comments to the Author” section, enter your conflict of interest statement in the “Confidential to Editor” section, and submit your "Accept" recommendation.

Reviewer #1: All comments have been addressed

Reviewer #2: All comments have been addressed

2. Is the manuscript technically sound, and do the data support the conclusions?

Reviewer #1: Yes

Reviewer #2: (No Response)

3. Has the statistical analysis been performed appropriately and rigorously? 

Reviewer #1: (No Response)

Reviewer #2: (No Response)

4. Have the authors made all data underlying the findings in their manuscript fully available?

Reviewer #1: (No Response)

Reviewer #2: (No Response)

5. Is the manuscript presented in an intelligible fashion and written in standard English?

Reviewer #1: (No Response)

Reviewer #2: (No Response)

6. Review Comments to the Author

Reviewer #1: (No Response)

Reviewer #2: (No Response)

7. PLOS authors have the option to publish the peer review history of their article (what does this mean?). If published, this will include your full peer review and any attached files.

Reviewer #1: No

Reviewer #2: No

---

## [Editor Report · Acceptance letter]

13 Jun 2023

PONE-D-22-35422R1 

COVID-19-related stigma among infected people in Sweden; psychometric properties and levels of stigma in two cohorts as measured by a COVID-19 stigma scale 

Dear Dr. Eriksson:

I'm pleased to inform you that your manuscript has been deemed suitable for publication in PLOS ONE. Congratulations! Your manuscript is now with our production department. 

Kind regards, 

on behalf of

Dr. Nabeel Al-Yateem 

Academic Editor

PLOS ONE